

# Cuticle hydrocarbons in saline aquatic beetles

María Botella-Cruz[1], Adrián Villastrigo[2], Susana Pallarés[1], Elena López-Gallego[3], Andrés Millán[1] and Josefa Velasco[1]

[1] Department of Ecology and Hydrology, University of Murcia, Spain
[2] Institute of Evolutionary Biology (CSIC-Universitat Pompeu Fabra), Barcelona, Spain
[3] Instituto Murciano de Investigación y Desarrollo Agrario y Alimentario (IMIDA), Murcia, Spain

## ABSTRACT

Hydrocarbons are the principal component of insect cuticle and play an important role in maintaining water balance. Cuticular impermeability could be an adaptative response to salinity and desiccation in aquatic insects; however, cuticular hydrocarbons have been poorly explored in this group and there are no previous data on saline species. We characterized cuticular hydrocarbons of adults and larvae of two saline aquatic beetles, namely *Nebrioporus baeticus* (Dytiscidae) and *Enochrus jesusarribasi* (Hydrophilidae), using a gas chromatograph coupled to a mass spectrometer. The CHC profile of adults of both species, characterized by a high abundance of branched alkanes and low of unsaturated alkenes, seems to be more similar to that of some terrestrial beetles (e.g., desert Tenebrionidae) compared with other aquatic Coleoptera (freshwater Dytiscidae). Adults of *E. jesusarribasi* had longer chain compounds than *N. baeticus*, in agreement with their higher resistance to salinity and desiccation. The more permeable cuticle of larvae was characterized by a lower diversity in compounds, shorter carbon chain length and a higher proportion of unsaturated hydrocarbons compared with that of the adults. These results suggest that osmotic stress on aquatic insects could exert a selection pressure on CHC profile similar to aridity in terrestrial species.

## INTRODUCTION

Maintaining water balance is critical for insects survival, especially in arid and semi-arid regions (*Addo-Bediako, Chown & Gaston, 2001*; *Gibbs, Fukuzato & Matzkin, 2003*). This is true not only for terrestrial, but also for aquatic species which may be periodically exposed to dry conditions during seasonal droughts and dispersal events. Insects in saline waters are also exposed to hyperosmotic stress which alters water and ionic homeostasis (*Bradley, 2009*). Therefore, saline water insects in arid regions are challenged with contrasting osmotic gradients from the aquatic and the aerial environment. Managing water loss under such stressful conditions is a critical problem for aquatic insects, as they are thought to be more permeable to water than their terrestrial counterparts (*Beament, 1961*).

Among the diverse ways to minimize water loss in terrestrial insects, the control of cuticle permeability is one the most important mechanisms (*Chung & Carroll, 2015*; *Rajpurohit et al., 2017*), but its role in aquatic ones has been less explored (e.g., *Jacob & Hansen, 1986*;

Corresponding author
María Botella-Cruz,
maria.botella1@um.es

*Alarie, Joly & Dennie, 1998*). The epicuticle of insects is covered with complex mixtures of nonpolar and polar compounds (*Gołębiowski et al., 2008*), being hydrocarbons the principal hydrophobic compounds of this layer, representing in some cases more than 90% of the cuticle (*Hadley, 1977*; *Maliński et al., 1986*). Insect cuticular hydrocarbons (CHCs) are thought to represent a primary adaptation to desiccation imposed by the transition to a terrestrial existence (*Jallon et al., 1997*). CHCs are exceptionally diverse and include complex mixtures of straight-chain compounds (n-alkanes), branched alkanes and unsaturated compounds (*Lockey, 1988*). Increases in the amount of CHCs or changes in their chemical composition resulting in increased chain length, linearity, and saturation are the main means of minimizing cuticular transpiration in insects (*Benoit, 2010*; *Gibbs & Rajpurohit, 2010*). Besides the key role of CHCs in preventing water loss (*Gibbs & Rajpurohit, 2010*; *Savković, Vučković & Stojković, 2012*), they are also involved in other important functions, such as protecting insects from microorganisms (*Stinziano et al., 2015*), chemical communication for recognition between closely related taxa (e.g., *Howard & Blomquist, 2005*; *Billeter et al., 2009*; *Savković, Vučković & Stojković, 2012*; *Pattanayak et al., 2014*; *Zhang et al., 2014*), sexual recognition (*Carlson et al., 1971*; *Jacob & Hansen, 1986*) or signalling of age and individual reproductive status (*Cuvillier-Hot et al., 2001*). CHCs with chain lengths ranging from approximately 21 to 50 carbons are usually related to cuticular permeability, while those with fewer than 21 carbons (volatile compounds) are involved in other functions (*Chung & Carroll, 2015*), such as pheromones or defensive compounds (*Blomquist & Bagnères, 2010*). Characterization of insect CHCs may therefore provide valuable information on many aspects of insect physiology and ecology.

CHC profiles are shaped by phylogenetic constraints; for example, CHCs in Coleoptera display common features at superfamily or family levels reflecting evolutionary tendencies (*Jacob & Hansen, 1986*). However, the amount and composition of CHCs also shows an important variation between species and populations reflecting local adaptation and it is strongly associated with desiccation resistance (*Gibbs, Chippindale & Rose, 1997*; *Kwan & Rundle, 2010*). For example, in desert Tenebrionids, the specific profile of CHC, characterized by high proportions of long chain lengths of branched alkanes, is thought to be a physiological adaptation to aridity (*Hadley, 1978*; *Lockey, 1980*). Similarly, CHC profiles varied predictably in populations of *Drosophila melanogaster* based on known associations between chain length, environmental variables and ecological function (*Rajpurohit et al., 2017*). In aquatic insects, salinity could exert a selective pressure on CHCs so that saline species could be expected to have a higher relative abundance of long-chain CHCs (higher impermeability) than freshwater ones. However, most studies on CHC composition and their functions have been carried out on terrestrial insects (e.g., *Blomquist & Jackson, 1979*; *Lockey, 1980*; *Alabi et al., 2011*; *Pattanayak et al., 2014*; *Stinziano et al., 2015*; *Rajpurohit et al., 2017*), whereas among aquatic ones, CHC profiles have only been described for some freshwater dytiscids (*Jacob & Hansen, 1986*; *Alarie, Joly & Dennie, 1998*). On the other hand, CHCs show a significant degree of plasticity conferring a notable intraspecific variability (*Howard & Blomquist, 1982*; *Gibbs & Rajpurohit, 2010*). Many studies have reported differences in CHC profiles within species depending on sex (e.g., *Beran et al., 2014*; *Pattanayak et al., 2014*), developmental stage (e.g., *Bagnères et al., 1996*), the

feeding state of individuals (e.g., *Jacob & Hansen, 1986*; *Alabi et al., 2011*), environmental conditions (*Toolson, 1982*) or rearing temperature (e.g., *Rouault et al., 2004*; *Rajpurohit et al., 2017*).

In inland saline waters, Coleoptera is one of the most representative and diverse insect orders (*Millán et al., 2011*) and, therefore, have been recently used as model organisms to study physiological tolerances to the main natural stressors in these systems, i.e., temperature, salinity and desiccation (e.g., *Sánchez-Fernández et al., 2010*; *Pallarés et al., 2012*; *Céspedes et al., 2013*; *Pallarés et al., 2016*). However, the potential role of cuticle permeability in driving stress tolerance in water beetles is unknown.

The aim of this study was to characterize CHC profiles of two saline water beetles representative of two of the most common families of Coleoptera in inland waters, *Nebrioporus baeticus* (Schaum) (family Dytiscidae, suborder Adephaga) and *Enochrus jesusarribasi* Arribas & Millán (family Hydrophiliade, suborder Polyphaga). Specifically, we address the following questions: (1) Do CHC profiles of saline water beetles show similar or different patterns to those found in other aquatic Coleoptera; (2) Do CHC profiles differ between the two studied species; and (3) Do CHCs show intraspecific variation in relation to sex and life stage within the studied species?

Because longer chain-length CHCs are thought to be more effective at preventing water loss (*Gibbs, 1998*), we expected a higher proportion of these compounds in (i) the two saline studied species compared with freshwater ones, (ii) the most halotolerant of the studied species (*E. jesusarribasi*, see 'Material and Methods'), (iii) adults compared with larvae in both studied species.

## MATERIALS AND METHODS

### Study species, specimens collection and maintenance

The studied species belong to two distant lineages of beetles (suborders Polyphaga and Adephaga) that have successfully colonized saline waters, showing a high osmoregulatory ability across a wide range of salinities (*Pallarés et al., 2015*). Adults of the most halotolerant species, *E. jesusarribasi,* are crawling, herbivorous and usually found in the shallow margins of hypersaline water bodies, while those from *N. baeticus* inhabit mesosaline waters and are active diving predators (*Millán et al., 2014*). Larvae of both species are benthic, carnivorous and desiccation-sensitive. Flying adults are the main source of colonizers during seasonal droughts.

Adults and larvae (second and third stages) specimens of *N. baeticus and E. jesusarribasi* were collected from typical localities in SE Spain in October 2015, where they constitute highly abundant populations, namely Chícamo stream (mean conductivity: 20 mS cm$^{-1}$) and Rambla Salada stream (mean conductivity: 84 mS cm$^{-1}$) with the collection permission number 201600150115 from the Consejeria de Agua, Agricultura y Medio Ambiente, Región de Murcia. Adults and larvae of each species were separately maintained in the laboratory for 48 h at 20 °C in 4 L aquaria containing water and substrate from the collection site.

## Extraction and analysis of cuticular hydrocarbons

Prior to CHC extraction, individuals of both life stages were killed by freezing at $-20\,°C$ in glass vials. CHCs of adult males ($n = 10$), females ($n = 10$) and larvae of each species ($n = 10$ for *N. baeticus* and $n = 16$ for *E. jesusarribasi*) were extracted individually in 2 mL vials by submerging each specimen into 175 µL of n-hexane containing 10 ng µL$^{-1}$ of octadecane ($C_{18}$) as an internal standard (e.g., *Kwan & Rundle, 2010*; *Stinziano et al., 2015*). Vials were continuously stirred for 5 h at 20 °C. The lipid extract was placed in borosilicate glass microinserts and evaporated and concentrated to dryness under a gentle stream of gas nitrogen. The residue was dissolved in 20 µL of hexane and ultrasonicated for 2 min (e.g., *Gołębiowski et al., 2011*; *Savković, Vučković & Stojković, 2012*). After CHC extraction, adults were sexed by examining genitalia in a stereomicroscope (Leica M165C with a Leica MEB10 fibre optic illuminator).

CHCs were identified and quantified by gas chromatography-mass spectrometry (GC-MS) using a 7890B GC system (Agilent Technologies, Santa Clara, CA, USA) and 5977 MSD (Network Mass selective Detector (MS) fitted with a HP-5 phenylmethyl siloxane column of 30 m × 250 µm × 0.25 µm a pulsed split less inlet (at 250 °C). The temperature program began at 70 °C, ramping at 30 °C min$^{-1}$ to 200 °C, slowing to 5 °C min$^{-1}$ to 310 °C, then ramping at 120 °C min$^{-1}$ to 310 °C and holding for 5 min.

The basic characterization of CHC structures was conducted by interpreting their EI mass spectra (number of carbons, methyl branching in saturated chains and double bonds in unsaturated chains). N-alkanes were identified by comparison of retention times with n-alkane standards (C10–C40; Sigma Aldrich, St. Louis, MO, USA). Branched alkanes and unsaturated compounds were identified by comparing the Kovats index (KI) with those of known compounds and by comparison of mass spectra using the NIST5 library.

Adjustments were made to peak time based on the time and area of the octadecane standard (e.g., *Arcaz et al., 2016*). Relative abundance of each CHC was expressed as the proportion of its adjusted peak area on the total adjusted areas (the sum of the adjusted areas of all the CHCs). The absolute amount of each compound was calculated according to the known amount of octadecane present within the sample based on the area under the GC peak. The amount of total CHCs of each specimen was then stimated as the sum of the abundance of all the CHCs.

## Data analysis

Inter and intraspecific differences on cuticular profiles were examined by means of Principal Components Analysis (PCA), performed with the R package FactoMineR. For adults, scores for the first three PCA factors were used as dependent variables in a multivariate analysis of variance (MANOVA) to test for differences in CHC profiles between species and sexes. The interaction term was included to assess whether sex-specific differences in CHC composition were consistent between the two species. CHC profiles of larvae were compared between species using ANOVA and the first PCA factor scores as the dependent variable. Differences in the relative abundance of the major classes of CHC compounds (i.e., n-alkanes, branched alkanes (methyl-alkanes and other branched alkanes) and unsaturated compounds) between species, stages and sexes were also assessed by ANOVAs.
Relative abundance data were arcsine square-root transformed for analyses. Normality and homocedasticity assumptions were validated on model residuals by graphical inspection (plots of residuals *versus* fitted values and Q–Q plots) (*Zuur et al., 2009*). Because CHCs ≤20C and CHCs >20C are involved in different biological functions, these analyses were made separately for each group. All statistical analyses were performed in R studio version 0.99.896.

## RESULTS

### Overall CHC profiles

The total number of CHCs in adults of *N. baeticus* was 57 for males and 50 for females. In *E. jesusarribasi*, 46 CHCs were identified in males and 56 in females. The longest chain length CHC of adults of *E. jesusarribasi* was hexatriacontane (36C), while that of *N. baeticus* was shorther (31C), corresponding to tritriacontane. Larvae of both species had a lower number of CHC compounds than adults (25 in *N. baeticus* and 20 in *E. jesusarribasi*) and the former had shorter chain lengths. CHC length of larvae of both species ranged from 14 to 24 carbon atoms, the longest CHC being an unidentified branched alkane in *N. baeticus* and an unidentified unsaturated CHC in *E. jesusarribasi* (see Table S1 for specific information of CHC compounds). The total amount of CHCs was also higher in adults of both species than larvae life stage (Table S1).

The most abundant CHC in adults was a brached alkane compound in both species. In *N. baeticus*, it was an undeterminated one (25C) in males and the 4-methyl pentacosane in females (25C), while in *E. jesusarribasi* it was n-dimethyl tritriacontane (33C) in both sexes. In larvae, the most abundant compound was docosene (22C) in *N. baeticus* and octadecene (18C) in *E. jesusarribasi*, both unsaturated CHCs (Table S1).

The PCA returned two principal factors that explained 32.65% and 9.94% of the total variance in adults and 46.83% and 12.23% in larvae. Two-dimensional ordination plots of PCA analysis showed a clear differentiation between CHC profiles in both species. The first factor divided samples by species both in adults and larvae stages (Fig. 1). The second factor separated adults by sexes, grouping females in the positive and males on the negative side of the axis (Fig. 1). The distribution pattern revealed larger differences in CHC composition between sexes in *N. baeticus* than in *E. jesusarribasi* as well as a higher intraspecific variability in larvae of the latter.

In adults, MANOVA analyses showed significant differences in CHC composition between species (Pillai's Trace = 0.99, $df = 33$, $p < 0.001$), sex (Pillai's Trace = 0.93, $df = 33$, $p < 0.001$) and their interaction (Pillai's Trace = 0.95, $df = 33$, $p < 0.001$), consistent with the patterns found by PCA. The compound that contributed most to the differentiation between species was tricosane (23C), only present in *N. baeticus* (Table 1). Methyl-alkane (27C) was the most contributing compound in the differentiation between sexes, being only present in *N. baeticus* females.

In larvae, CHC profiles significantly differed between species ($F = 554.3$, $p < 0.001$). The compound that contributed most to such differentiation was an unsaturated compound (22C), which was ten times more abundant in *N. baeticus* than in *E. jesusarribasi*.

**A**

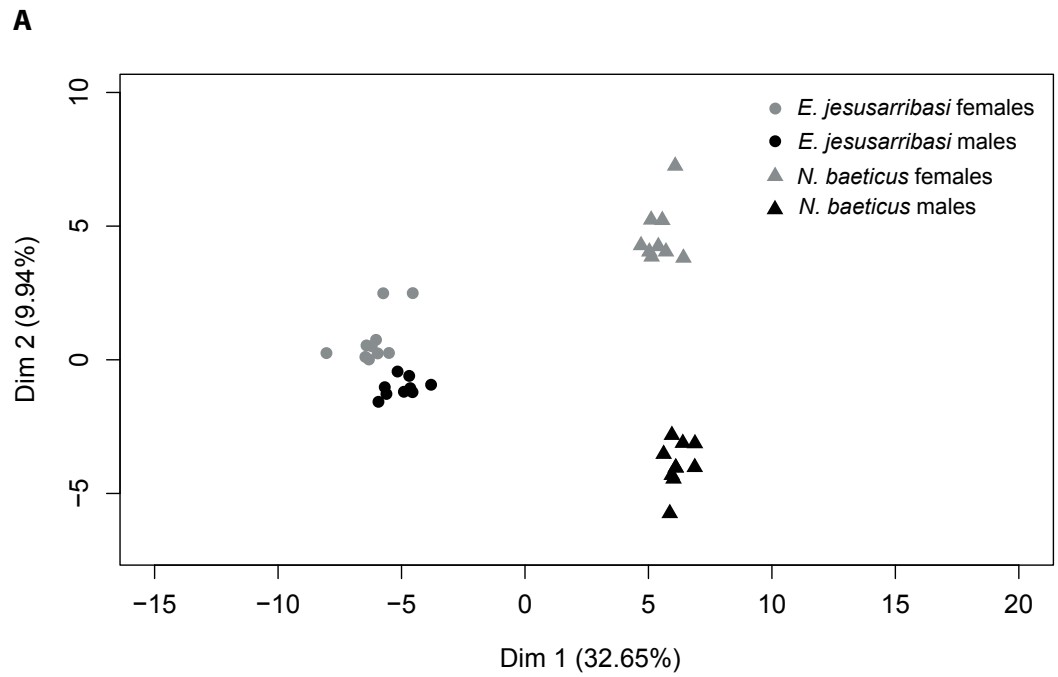

**B**

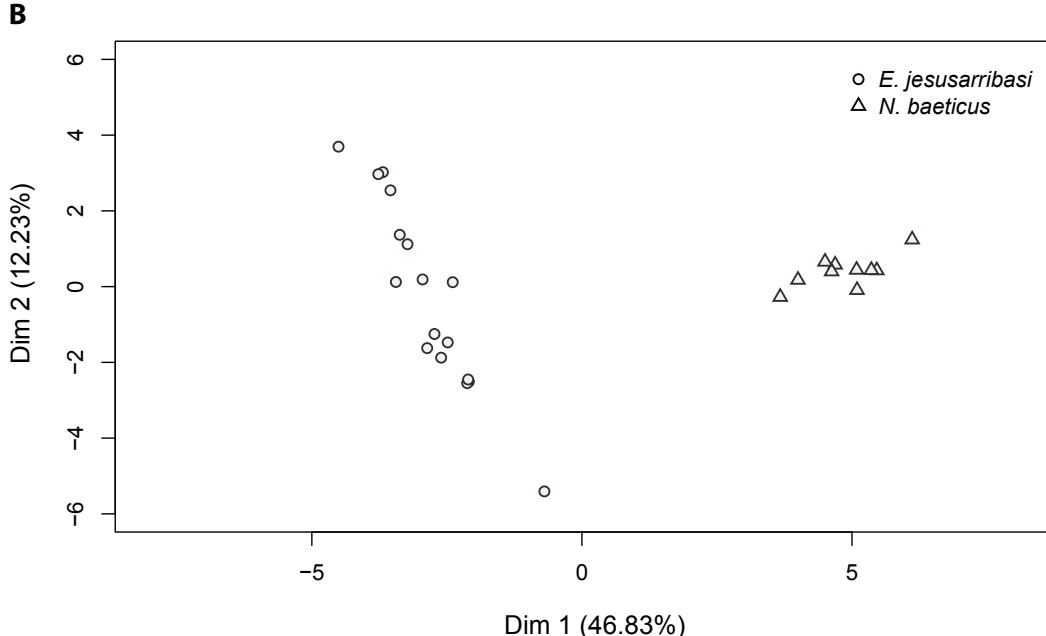

**Figure 1** Projection of principal component analysis (PCA) factor scores with the first two PCA factors of the quantitative patterns of cuticular hydrocarbons (CHCs) in adults (A) and larvae (B) of *Nebrioporus baeticus* and *Enochrus jesusarribasi.*

**Table 1 Total number of cuticle hydrocarbons compounds and relative abundances of the main classes for adults (A), females (F), males (M) and larvae (L) life stages of the studied species.** CHCs were analyzed in two separated groups in function of its chain length ($\leq$20 C and >20 C).

| Species | Life stage | Sex | Total | Alkanes | | | | | | Unsaturated | |
| | | | | n-Alkanes | | Methyl branched | | Other branched | | | |
| | | | | n° | % | n° | % | n° | % | n° | % |
|---|---|---|---|---|---|---|---|---|---|---|---|
| | | | | CHCs $\leq$ 20C | | | | | | | |
| | A | F | 7 | 4 | 48.04 | 0 | 0 | 1 | 1.96 | 2 | 50.00 |
| N. baeticus | A | M | 7 | 4 | 54.33 | 0 | 0 | 0 | 0 | 3 | 45.66 |
| | L | – | 10 | 3 | 14.86 | 1 | 2.7 | 3 | 8.22 | 3 | 74.22 |
| | A | F | 11 | 4 | 45.97 | 0 | 0 | 4 | 11.88 | 3 | 42.15 |
| E. jesusarribasi | A | M | 3 | 3 | 100 | 0 | 0 | 0 | 0 | 0 | 0 |
| | L | – | 11 | 5 | 20.81 | 0 | 0 | 2 | 7.2 | 4 | 71.99 |
| | | | | CHCs > 20C | | | | | | | |
| | A | F | 43 | 11 | 25.37 | 17 | 51 | 5 | 17.28 | 10 | 6.35 |
| N. baeticus | A | M | 50 | 13 | 18.22 | 17 | 42.51 | 10 | 34.17 | 10 | 5.10 |
| | L | – | 15 | 2 | 8.57 | 3 | 5.62 | 2 | 1.34 | 8 | 84.47 |
| | A | F | 45 | 8 | 16.04 | 13 | 43.32 | 16 | 35.46 | 8 | 5.18 |
| E. jesusarribasi | A | M | 43 | 7 | 13.05 | 19 | 47.09 | 10 | 32.31 | 7 | 7.55 |
| | L | – | 9 | 0 | 0 | 0 | 0 | 0 | 0 | 9 | 100 |

**Table 2 Comparison of the relative abundance (%) of the main cuticle hydrocarbons classes of the study species with freshwater beetles.**

| Habitat | Family | Species | Sex | ALL CHCs | | | | | |
| | | | | Alkanes | | | Unsaturated | Unidentified | |
| | | | | n-Alkanes | Methyl branched | Other branched | | | |
|---|---|---|---|---|---|---|---|---|---|
| Hypersaline | Hydrophilidae | E. jesusarribasi | F | 17.64 | 41.3 | 34.00 | 7.11 | | Present study |
| | | | M | 13.38 | 46.83 | 32.38 | 7.41 | | ,, |
| Mesosaline | Dytiscidae | N. baeticus | F | 26.47 | 59.47 | 16.54 | 8.48 | | ,, |
| | | | M | 19.34 | 41.9 | 32.36 | 6.40 | | ,, |
| Freshwater | Dytiscidae | Agabus anthracinus | – | 46.9 | 25.9 | 0 | 27.1 | | Alaire et al. (1998) |
| Freshwater | Dytiscidae | Agabus bipustulatus | – | 52.7 | 0 | 0 | 47.3 | | Jacob & Hansen (1986) |
| Freshwater | Dytiscidae | Dytiscus marginalis | F | 78.5 | 3.4 | 5.6 | 8.3 | 4.2 | ,, |
| | | | M | 36 | 2.7 | 1.8 | 59.5 | | ,, |
| Freshwater | Dytiscidae | Ilybius angustior | M | 43 | 1.5 | 1.8 | 51.6 | 2.1 | ,, |

## Patterns in CHC classes

In general, methyl-alkanes were the most abundant class of CHCs in adults of both species, representing between 41–59% of the total CHCs (Tables 1 and 2), while unsaturated compounds were the dominant class in larvae (>80%). Significant differences in abundance of all CHC classes were found between larvae and adults in the two studied species, both in CHCs $\leq$20C and CHCs >20C (Table S2).

**Table 3  Species, sex and its interaction effects on the relative abundance of the main cuticle hydrocarbons (CHCs ≤ 20C) in adults.**

| Class | | df | F value | P value |
|---|---|---|---|---|
| n-Alkanes | Species | 1 | 11.92 | 0.001 |
| | Sex | 1 | 26.80 | <0.001 |
| | Species*Sex | 1 | 13.30 | <0.001 |
| | Residuals | 34 | | |
| Unsaturated | Species | 1 | 51.23 | <0.001 |
| | Sex | 1 | 94.11 | <0.001 |
| | Species*Sex | 1 | 36.49 | <0.001 |
| | Residuals | 34 | | |

**Notes.**
*df*, degrees of freedom.

**Table 4  Species, sex and its interaction effects on the relative abundance of the main cuticle hydrocarbons classes (CHCs > 20C) in adults.**

| Class | | df | F value | P value |
|---|---|---|---|---|
| n-Alkanes | Species | 1 | 28.88 | <0.001 |
| | Sex | 1 | 13.01 | <0.001 |
| | Species*Sex | 1 | 2.67 | 0.12 |
| | Residuals | 34 | | |
| Branched alkanes (Methyl-alkanes and others) | Species | 1 | 19.15 | <0.001 |
| | Sex | 1 | 17.97 | <0.001 |
| | Species*Sex | 1 | 1.89 | 0.17 |
| | Residuals | 34 | | |
| Unsaturated | Species | 1 | 0.12 | 0.73 |
| | Sex | 1 | 0.11 | 0.74 |
| | Species*Sex | 1 | 1.74 | 0.19 |
| | Residuals | 34 | | |

**Notes.**
*df*, degrees of freedom.

## Compounds shorter than 20 carbon atoms

Volatile CHCs were represented almost equally by n-alkanes and unsaturated compounds in females of both species and in males of *N. baeticus*. In *E. jesusarribasi* males, the n-alkanes represented the 100% of CHCs (Table 3). Methyl-alkanes were absent in both species. Despite these similar abundance patterns, the relative abundance of unsaturated and n-alkane compounds significantly differed between species, being higher in *N. baeticus* than in *E. jesusarribasi* (Table 4). The relative abundance of these classes also differed between sexes showing a contrasting pattern on each species (i.e., significant species × sex interaction, see Table 4). In larvae, significant differences were also found between species in relative abundance of n-alkanes ($F = 43.27$, $p < 0.001$) and unsaturated compounds ($F = 37.63$, $p < 0.001$).

### Compounds longer than 20 carbon atoms

Branched alkanes, especially methyl-alkanes, was the most abundant class in adults of both species, followed by n-alkanes and unsaturated compounds (Table 3). Significant differences in relative abundance of n- alkanes and branched alkenes were found between species and sexes (Table 4). *Nebrioporus baeticus* showed a higher abundance of n-alkanes compared with *E. jesusarribasi* and females of both species showed a significantly higher abundance than males. The opposite patterns were found for branched alkanes. In larvae, unsaturated compounds represented 84% in *N. baeticus* and 100% in *E. jesusarribasi* (Table 3), being this difference in abundance highly significant ($F = 8.60$, $p < 0.01$).

## DISCUSSION

The CHC profile characterized for adults of the two species of saline water beetles studied here differed from that of other freshwater beetles, and showed common patterns to those generally attributed to adaptation to aridity in terrestrial Coleoptera. This points to an important role of cuticle permeability in driving tolerance to salinity and desiccation in these species. Comparison of CHC profiles between adults of *N. baeticus* and *E. jesusarribasi* and between life-stages and sexes within each species also revealed potential inter and intraspecific differences in cuticle permeability likely related with differences in tolerance to osmotic stress.

### Interspecific variation in CHCs

We found marked differences in the patterns of CHC profiles between the saline studied species and those previously reported for freshwater ones (*Jacob & Hansen, 1986*; *Alarie, Joly & Dennie, 1998*). In particular, if the CHC composition of *N. baeticus* is compared with that of other freshwater species within the family Dytiscidae (Table 2), the cuticle of the saline species was characterized by a higher abundance of longer chain branched alkanes, while freshwater species display a relatively complex spectrum of CHCs with predominating amounts of unbranched components (n-alkanes and unsaturated alkenes) (Table 2). Methyl-alkanes melt 10–30 °C below n-alkanes with the same chain length, depending on the location of the methyl branch (*Gibbs & Pomonis, 1995*). The abundance of long-chain branched compounds and their interactions with other alkanes compounds will determine the overall waterproofing properties of the surface lipids. Accordingly, the CHC profile of the saline beetles studied here, dominated by more complex compounds, is expected to confer them a more impermeable cuticle than that of freshwater ones. This is likely an adaptation of insects living in temporary saline waters in arid climatic regions to the osmotic stress imposed by water salinity and desiccation during seasonal droughts. Previous studies have reported differences in water loss rates between beetle species with different saline optima (*Pallarés et al., 2016*) or between freshwater and saline populations of corixids (*Cannings, 1981*), supporting such hypothesis. Furthermore, a recent transcriptomic study in *Anopheles* larvae has suggested that cuticle composition may be altered to deal with osmoregulatory stress by decreasing permeability in saline water, as cuticle and cytoskeleton genes were robustly induced at 40–50% seawater salinities (*Uyhelji, Cheng & Besansky, 2016*).

Some of the characteristics of CHC profiles described in the saline species studied here have been also shown in terrestrial beetles adapted to aridity (*Jacob & Hansen, 1986*; *Lockey, 1979*; *Lockey, 1988*; *Nelson & Charlet, 2003*). For example, the predominant class of CHCs in desert Tenebrionidae, with an exceptionally thick and impermeable epicuticular wax layer, are branched alkanes (*Crowson, 1981*), like in adults of the two studied species. In five desert species from Arizona, no unsaturated hydrocarbons were detected in the cuticle (*Jacob & Hansen, 1986*) and the alkanes included both straight and branched chains, having the latter generally more carbon atoms (*Crowson, 1981*). In the tenebrionid beetle *Eleodes armata* LeConte, *1851* and a house cricket, *Acheta domesticus* L., 92% of the branched compounds were alkanes (*Jackson & Blomquist, 1976*; *Hadley, 1977*). Thus, salinity could impose a selective pressure on CHC profile of aquatic insects similar to that exerted by aridity in terrestrial species. Long-chain methylbranched hydrocarbons could have an important role in limiting water loss by osmosis or by transpiration through the cuticle.

The association between salinity, desiccation and CHC composition is also supported if the CHCs of the two studied species are compared. A similar total number of compounds was identified in both species suggesting a similar complexity of cuticle chemistry, but differences in chain length and specific CHCs were found, pointing to a more impermeable cuticle in *E. jesusarribasi* than in *N. baeticus.* Carbon chain length of CHCs ranged up to 36C in *E. jesusarribasi* and 31C in *N. baeticus.* In addition, most of the compounds of *E. jesusarribasi* ranged between 31–36C chain length. A high percentage of long-chain hydrocarbons has been shown to confer impermeability to the cuticle in other arthropods (e.g., *Hadley, 1977*; *Toolson & Hadley, 1977*; *Lockey, 1980*; *Gibbs & Pomonis, 1995*; *Gibbs, Fukuzato & Matzkin, 2003*; *Gibbs & Rajpurohit, 2010*). The contribution of CHCs in driving differences in stress tolerance between aquatic beetles needs to be further investigated, but the differences in cuticle permeability between the two species inferred from our results are consistent with the higher desiccation resistance (*Pallarés et al., 2017*), osmoregulatory ability and salinity tolerance (*Pallarés et al., 2015*) of *E. jesusarribasi* compared to *N. baeticus.* Specifically, the average water loss rates under desiccation conditions (40% RH) was 4.04% of fresh mass h$^{-1}$ in *N. baeticus* and 1.58% of fresh mass h$^{-1}$ in *E. jesusarribasi* (*Pallarés et al., 2017*).

## Intraspecific variation in CHCs

The different CHC profiles between larvae and adults within the two studied species were also consistent with the expected differences in cuticle permeability between mature and immature stages. Larvae had a remarkably lower number of CHCs with shorter chain length compared with adults in both species. Furthermore, unsaturated compounds were the most abundant CHC class in larvae, as expected according to their thinner, softer and more permeable cuticle if compared with adults, and therefore less effective against water loss (*Chapman, 1975*). Adults showed a lower abundance of unsaturated CHCs and a greater concentration of branched (their most abundant CHC class) than n-alkanes. These compounds, with higher molecular weight and melting temperatures (*Gibbs & Pomonis, 1995*), may confer adults cuticle a higher resistance to water loss (*Chung & Carroll, 2015*), as required during flight dispersal.

Such differences in CHC complexity between adults and larvae reveal an important ontogenetic modification of the cuticular lipids composition, in which chemical signature becomes enriched as the individual is developing to adult, with the increase of long-chain compounds with higher molecular weight. The main changes in CHC composition occur during the development from larval to adult stages, although sex dependent compositions also reflect a possible pheromonal function of CHCs $\leq$ 20C, usually carried out by volatile compounds (*Jacob & Hansen, 1986*).

The CHC profile described here for two saline water beetles suggests that the cuticle of aquatic coleopteran could have an important role in adaptation to salinity and desiccation. Studies comparing cuticular lipids and water loss rates among related water beetle species would provide a better understanding of how changes in lipid composition modulate cuticular transpiration in these insects. The relationship between CHC composition and salinity tolerance also needs to be further explored by comparison of CHC profiles between freshwater and saline species across beetle lineages and the study of the plasticity of cuticle permeability in relation with changes in salinity.

## ACKNOWLEDGEMENTS

We thank José Rodríguez and technicians from the SAI (Servicio de Apoyo a la Investigación de la Universidad de Murcia) for their technical support for GC-MS analysis. Thanks to María José Jordán (IMIDA) for her advice in CHCs identification. We also thank Yves Alarie and one anonymous referee for a constructive revision of our manuscript.

### Funding

This research was funded by the I+D+i project CGL2013-48950-C2-2-P (Spanish Ministry of Economy and Competitivity) and co-funded with Feder Funds. The funders had no role in study design, data collection and analysis, decision to publish, or preparation of the manuscript.

### Grant Disclosures

The following grant information was disclosed by the authors:
I+D+i project CGL2013-48950-C2-2-P (Spanish Ministry of Economy and Competitivity).
Feder Funds.

### Competing Interests

The authors declare there are no competing interests.

### Author Contributions

- María Botella-Cruz conceived and designed the experiments, performed the experiments, analyzed the data, contributed reagents/materials/analysis tools, wrote the paper, prepared figures and/or tables, reviewed drafts of the paper.
- Adrián Villastrigo performed the experiments, reviewed drafts of the paper.

- Susana Pallarés conceived and designed the experiments, analyzed the data, contributed reagents/materials/analysis tools, prepared figures and/or tables, reviewed drafts of the paper.
- Elena López-Gallego contributed reagents/materials/analysis tools, reviewed drafts of the paper, hydrocarbons identification.
- Andrés Millán conceived and designed the experiments, reviewed drafts of the paper.
- Josefa Velasco conceived and designed the experiments, performed the experiments, analyzed the data, contributed reagents/materials/analysis tools, reviewed drafts of the paper.

### Field Study Permissions

The following information was supplied relating to field study approvals (i.e., approving body and any reference numbers):

Specimens were collected under collection permission number 201600150115 from the Consejeria de Agua, Agricultura y Medio Ambiente, Región de Murcia.

### Data Availability

The raw data has been uploaded as a Supplemental File.

### Supplemental Information

Supplemental information for this article can be found online at http://dx.doi.org/10.7717/peerj.3562#supplemental-information.

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
