# Peer review of "Cuticle hydrocarbons in saline aquatic beetles"

_PeerJ, doi:10.7717/peerj.3562_

## Round 0.1 · original submission · Minor Revisions

· Academic Editor

Minor Revisions

Two colleagues have reviewed your manuscript and both of them, as well as myself, have much appreciated you work. The reviewers also underlined some minor points that need your attention.

Reviewer 1 ·

Basic reporting

Good

Experimental design

Good

Validity of the findings

Good

Comments for the author

I previously reviewed this manuscript for another journal. I thought the work was done well but too descriptive for that particular journal. The authors have done a very good job of responding to my previous comments, with one exception. An octadecane standard was included when CHCs were extracted, thereby allowing total CHC abundance and the absolute amount of each CHC to be determined. At the very least, the authors should consider including a table containing total CHC amounts for each species and life stage. This will allow interested readers to calculate absolute CHC amounts for themselves if they want to.

·

Basic reporting

Despite their diversity, aquatic Coleoptera cuticular hydrocarbons have yet to be studied in much detail. Whereas this paper adds documenting the diversity of cuticular hydrocarbons within members of aquatic Coleoptera, which is certainly useful, its foremost interest is to document for the first time species adapted to saline aquatic biota. The findings providing in this paper and the comparisons made with both freshwater and terrestrial beetle species whereas somewhat expected are fascinating and open to further research. It is with great delight that I have reviewed this manuscript, which was written in excellent English. The literature pertaining to the subject was extensively reviewed; the methodology used was adequate and the statistical analyses conducted were adequate. If I had only a suggestion to make it would be to perhaps provide a clearer time frame during which the specimens studied were collected. Indeed the cuticular hydrocarbon composition may vary considerably over time and for that reason only, some precision about the sampling time could prove to be important. I also want to point a minior error in the reference @ line 332: "Candian Entomology"should read "Canadian Entomologist" In concluding I need to highlight once again the high quality of this study.

Experimental design

cf above

Validity of the findings

The correlation made between salinity and terrestrial beetle species whereas expected is made for the first time, which opens to further interesting questions.

Comments for the author

Very good article!

---

## Round 0.2 · accepted · Accept

· Academic Editor

Accept

Thank you very much for improving the manuscript. Congratulations for your nice work.